# Change Detection Based on Fusion Difference Image and Multi-Scale Morphological Reconstruction for SAR Images

**Jiayu Xuan** [1] , **Zhihui Xin** [1,*], **Guisheng Liao** [2], **Penghui Huang** [3], **Zhixu Wang** [1] **and Yu Sun** [1]

1. School of Physics and Electronic Information Technology, Yunnan Normal University, Kunming 650500, China; xuanjiayu@user.ynnu.edu.cn (J.X.); wangzhixu@user.ynnu.edu.cn (Z.W.); sunyu@user.ynnu.edu.cn (Y.S.)
2. National Laboratory of Radar Signal Processing, Xidian University, Xi'an 710071, China; liaogs@xidian.edu.cn
3. School of Electronic Information and Electrical Engineering, Shanghai Jiao Tong University, Shanghai 200240, China; huangpenghui@sjtu.edu.cn
* Correspondence: xinzhihui@ynnu.edu.cn

**Abstract:** Synthetic aperture radar (SAR) image-change detection is widely used in various fields, such as environmental monitoring and ecological monitoring. There is too much noise and insufficient information utilization, which make the results of change detection inaccurate. Thus, we propose an SAR image-change-detection method based on multiplicative fusion difference image (DI), saliency detection (SD), multi-scale morphological reconstruction (MSMR), and fuzzy c-means (FCM) clustering. Firstly, a new fusion DI method is proposed by multiplying the ratio (R) method based on the ratio of the image before and after the change and the mean ratio (MR) method based on the ratio of the image neighborhood mean value. The new DI operator ratio–mean ratio (RMR) enlarges the characteristics of unchanged areas and changed areas. Secondly, saliency detection is used in DI, which is conducive to the subsequent sub-area processing. Thirdly, we propose an improved FCM clustering-change-detection method based on MSMR. The proposed method has high computational efficiency, and the neighborhood information obtained by morphological reconstruction is fully used. Six real SAR data sets are used in different experiments to demonstrate the effectiveness of the proposed saliency ratio–mean ratio with multi-scale morphological reconstruction fuzzy c-means (SRMR-MSMRFCM). Finally, four classical noise-sensitive methods are used to detect our DI method and demonstrate the strong denoising and detail-preserving ability.

**Keywords:** change detection; clustering; fusion difference image; morphological reconstruction; saliency detection; SAR image

## 1. Introduction

The goal of synthetic aperture radar (SAR) image-change detection is to generate a change image, which describes the changes of two or more time-phase images between different calibration times [1–3]. As SAR has imaging capability for all day and for all weather [4,5], it is widely used in environmental monitoring, ecological monitoring, urban development research, agriculture forestry monitoring, natural disaster assessment, and other fields [6–9].

The process of SAR image-change detection generally includes three parts: image preprocessing, difference image (DI) generation, and difference image analysis [10,11].

The first step mainly includes image registration and image denoising. Due to the large amount of speckle noise in SAR images, many denoising methods are used to improve the detection effect, such as Lee filtering [12], Frost filtering [13], non-local means (NLM) [14], and Speckle reducing anisotropic diffusion (SRAD) [15].

DI generation is an important step in change detection. The first type is based on the pixel. Dekker proposed the ratio (R) method [16]. The R method reduces the noise, but exaggerates the change degree of the low gray pixel area. Bazi et al. proposed the log-ratio (LR) method [17]. The method transforms the multiplicative noise into additive

noise, which is more conducive to the subsequent denoising work. The second type is based on neighborhood information such as mean value, median value, local variance, and weighted spatial distance. Inglada et al. proposed the mean-ratio (MR) method [18]. The MR operator can reduce the image noise. However, it will reduce the contrast of the changed area. Zheng et al. proposed a new operator by weighted fusion of the subtraction operator and LR operator [19]. Gong et al. proposed the neighborhood-based ratio (NR) method [20]. The NR operator makes full use of the spatial information of the image, but it will enlarge the gray level of the edge. Zhang et al. proposed a method based on super-pixel segmentation [21], which makes better use of neighborhood information, while maintaining image contours. Wang et al. proposed a DI-generation method based on the coefficient of variation and physical proximity [22]. Zhang et al. proposed a DI-generation method based on the adaptive generalized likelihood ratio test (AGLRT) [23]. This method greatly suppresses the noise in the image. Jia et al. fused the subtraction DI and ratio DI by multi-scale wavelet fusion [24].

The last step is DI analysis. Threshold and clustering are the most common unsupervised methods. Kittler and Illingworth (KI), Otsu method, and the expectation maximization (EM) method are widely used in change detection [25–27]. Clustering methods mainly include K-Means [28] clustering and fuzzy c-means clustering (FCM) [29].

FCM is the most widely used method. However, neighborhood information is not used in FCM, as it is very sensitive to noise, so the final segmentation effect is not ideal. Ahmed et al. proposed the offset corrected fuzzy c-means clustering (FCM_S) [30] method. Aiming at the low efficiency of FCM_S, Chen et al. proposed the improved FCM_S1 and FCM_S2 [31]. These two methods all use neighborhood information, but many experimental parameters need to be adjusted to get optimal results. Gong et al. used the improved fuzzy local information c-means (FLICM) method in SAR image-change detection [32]. This method has few parameters. However, the convergence speed is slow. Mu et al. proposed the fuzzy c-means clustering method based on the Gaussian kernel (KFCM) [33]. This algorithm can improve the extraction of image features but has high requirements for the selection of initial clustering centers. Wang et al. proposed the fuzzy adaptive local and region-level information c-means (FALRCM) [34]. Neighborhood information is adaptively utilized, and it is robust to noise. Lei et al. proposed the fast and robust fuzzy c-means (FRFCM) [35], which introduced the morphological images into fuzzy clustering. In deep learning, FCM is often used for image pre-classification. Gong et al. used FCM to pre-classify the DI and obtained reliable changed samples and unchanged samples. These samples are used to train Deep Neural Networks (DNN) [36].

Now, machine learning and deep learning are widely used in change detection. Gao et al. combined the neighborhood-based ratio and extreme learning machine (NR-ELM) [37], and Cui et al. proposed an unsupervised SAR change detection method based on stochastic subspace ensemble learning, which combined the training samples generated by two DIs [38]. Ma et al. proposed a method based on multi-grained cascade forest and multi-scale fusion [39]. Gao et al. proposed the convolution wavelet neural network (CWNN) [40] and the principal component analysis network (PCANet) [41]. Wang et al. proposed a novel multi-scale average pooling (MSAP) network to exploit the changed information from the noisy difference image [42]. Qu et al. proposed a dual-domain network (DDNet) [43], which introduced discrete cosine transform (DCT) to the net. However, machine learning and deep learning methods usually need a long running time and depend on the accuracy of labels.

In the existing DI methods, only single pixel or neighborhood information is used in these methods. Besides, the calculation of clustering is complex when using neighborhood information. The proposed saliency ratio–mean ratio with multi-scale morphological reconstruction fuzzy c-means (SRMR-MSMRFCM) has the following advantages. R operator and MR operator are used to generate fusion DI in our method. Therefore, the advantages of single pixel operator and neighborhood operator are combined in our method. Saliency detection can effectively extract the changed areas and unchanged areas, which lays the

foundation for the subsequent sub-regional processing. Moreover, neighborhood information is introduced through morphological reconstruction rather than directly adding fuzzy factors, which simplifies the operation.

This paper's remaining frame is organized as follows. The proposed method is described in Section 2 in detail. The experimental results on six data sets are shown in Section 3. Parameter analysis is shown in Section 4. The conclusion and further research direction are put forward in Section 5. The main contributions of this paper are as follows.

(1) A new difference image generation method is proposed. The R method and the MR method are combined by multiplication. In the case of preserving the details of the image, the features of the changed areas are effectively enlarged, and the features of the unchanged areas are suppressed.

(2) Saliency detection is used to obtain the changed and unchanged areas of the image. Large-size structuring elements are used to remove noise in the unchanged area. In the changed area, multi-scale morphological reconstruction can not only maintain the details of the image but also effectively remove the noise.

(3) FCM, Kmeans, Otsu, and manual threshold are very sensitive to noise. Though these methods are applied in the proposed method, the proposed method can decrease the influence of the noise and preserve the detail of the changed area better.

## 2. Materials and Methods

In this part, we introduce the proposed SAR image-change detection method in detail. This method can be divided into the following steps: difference image generation, saliency detection, sub-regional morphological reconstruction, and output detection results.

Firstly, the R operator and the MR operator are used to generate DI, and the two images are fused into a new DI by multiplication. Secondly, saliency detection is used for DI, and Otsu method is used to obtain the binary saliency image. Thirdly, according to the saliency image, we reconstruct the image by sub-regional morphology. Finally, FCM is used to output the change detection result. Figure 1 is the flow chart of the method in this paper. In the proposed method, the new DI operator effectively increases the contrast between the changed and unchanged areas and increases the accuracy of saliency detection. Saliency detection is the basis of sub-regional morphological reconstruction. The combination of the two not only removes noise, but also preserves image details. By introducing the morphological reconstruction image information into FCM, FCM also has strong robustness to noise. The details of DI generation, saliency detection, multi-scale morphological reconstruction, and FCM are reported in Sections 2.1–2.4

### 2.1. Generation of Difference Image

The ratio method reduces the influence of multiplicative noise and increases the contrast of the changed area. However, the additive noise generated by this method still exists in large quantities. Firstly, the normalized ratio method is used to obtain the initial DI. Compared with the original ratio method, this method reduces the weight of the change difference for low gray pixels, while the weight of the change difference for high gray pixels is almost unchanged. This improves the accuracy. Assuming that images $T_1$ and $T_2$ are SAR images of different times in the same area, the difference image of $T_1$ and $T_2$ can be obtained by Equation (1).

$$X_{d1}(x,y) = \frac{\max\{T_1(x,y), T_2(x,y)\} - \min\{T_1(x,y), T_2(x,y)\}}{\max\{T_1(x,y), T_2(x,y)\} + \min\{T_1(x,y), T_2(x,y)\}} \tag{1}$$

The MR method is to take the neighborhood mean of the corresponding pixels and then calculate the ratio. It is strongly robust to scatter noise. The MR method can be calculated by Equation (2).

$$X_{d2}(x,y) = 1 - \min\left\{\frac{\mu_1(x,y)}{\mu_2(x,y)}, \frac{\mu_2(x,y)}{\mu_1(x,y)}\right\} \tag{2}$$

where $\mu_1(x,y)$ and $\mu_2(x,y)$ are the average of the gray values of all pixels in the $3 \times 3$ neighborhood window centered on coordinate $(x,y)$ in images $T_1$ and $T_2$, respectively.

In this paper, the R operator and the MR operator are multiplied and normalized to form a new DI operator. The ratio–mean ratio (RMR) DI can be calculated by Equation (3). The normalized image is shown by Equation (4).

$$X_{d3} = \frac{\max\{T_1(x,y),T_2(x,y)\}-\min\{T_1(x,y),T_2(x,y)\}}{\max\{T_1(x,y),T_2(x,y)\}+\min\{T_1(x,y),T_2(x,y)\}} \\ \times \left\{1 - \min\left\{\frac{\mu_1(x,y)}{\mu_2(x,y)}, \frac{\mu_2(x,y)}{\mu_1(x,y)}\right\}\right\} \tag{3}$$

$$X_{d4} = \frac{X_{d3}(x,y) - \min(X_{d3})}{\max(X_{d3}) - \min(X_{d3})} \tag{4}$$

When a single pixel and its neighborhood change greatly, the gray value of the new operator will be still large after normalization. When the change of a single pixel and its neighborhood change little, the gray value of the new operator will be very small after normalization. Therefore, the contrast between the changed area and the unchanged area is improved. In addition, the proposed RMR method reduces the false negative caused by R operator and the false positive caused by MR operator through multiplication operator.

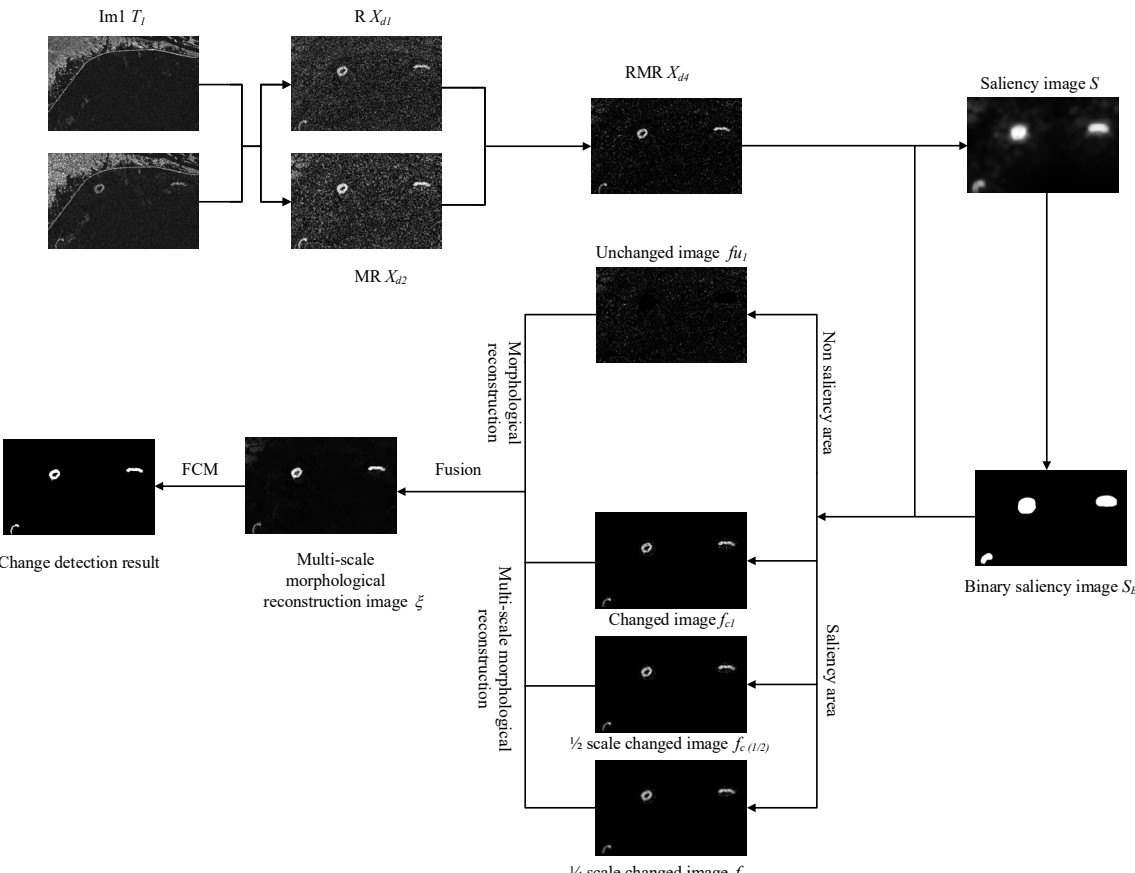

**Figure 1.** Flowchart of the proposed method.

### 2.2. Saliency Detection

Saliency detection (SD) is suitable for SAR image-change detection. In paper [44,45], saliency detection was applied to select training samples. Context-Aware (CA) was proposed by Goferman et al. in 2012 [46]. The author thought that the saliency image should contain both the target and the background area near the target. In this way, the salient information of the image can be better described. Therefore, the changed area must be in the salient area. In this paper, the RMR DI is used for saliency detection. The principle of CA is as follows. RMR DI is converted from RGB space to Lab space. The distance between RMR DI blocks is obtained by the following.

$$d(p_i, p_j) = \frac{d_{color}(p_i, p_j)}{1 + c \cdot d_{position}(p_i, p_j)} \tag{5}$$

where $d_{color}(p_i, p_j)$ is the Euclidean distance of color between area $p_i$ and $p_j$, and $d_{position}(p_i, p_j)$ is the Euclidean distance of space. Therefore, the distance between $p_i$ and $p_j$ is proportional to the color distance and inversely proportional to the spatial distance. Here $c$ is set to 3. Then, the saliency value can be calculated according to the distance.

$$S_i^r = 1 - \exp\left(-\frac{1}{K}\sum_{K=1}^{K} d(p_i, p_k)\right) \tag{6}$$

where $S_i^r$ is the saliency value. At scale $r$, the image is segmented into $K$ areas. The greater the dissimilarity calculated by the area, the greater the saliency value of the pixels in the area. Here $K$ is set to 64. In order to enhance the contrast between salient areas and non-salient areas, the above calculation is extended to multi-scale. When a pixel has a large saliency value at multiple scales, it is considered as the salient area we are looking for. Therefore, the introduction of the mean saliency value is necessary, which is calculated by the following.

$$\overline{S_i} = \frac{1}{M}\sum_{r \in R} S_i^r \tag{7}$$

where $R = \{r_1, \ldots r_M\}$ is the set of scales of the area, and $\overline{S_i}$ is the average saliency value of the area at these scales. Here $R$ is set to $\{100\%, 80\%, 50\%, 30\%\}$. Extract the most focused local areas at each scale from the saliency image, and a pixel is considered to be of attention at the current scale if its saliency value exceeds a certain threshold. The saliency value is refined as

$$\hat{S}_i = \overline{S_i} \cdot \left(1 - d_{foci}^r(i)\right) \tag{8}$$

where $d_{foci}^r(i)$ is the distance between area $i$ and the nearest area of attention. Assuming that the image obeys a two-dimensional Gaussian distribution, the final saliency value $S$ of the image can be calculated as

$$S = \hat{S}_i \cdot G \tag{9}$$

where $G$ is a two-dimensional Gaussian distribution at the center of the image. After obtaining the saliency image, the image is binarized in order to more intuitively reflect the changed area and unchanged area. Otsu [26] method can effectively distinguish the background from the target. Therefore, it is used to segment the image. Then, we get the binary saliency image $S_B$. In $S_B$, the white part is the general changed area, and the black part is the general unchanged area. Through $S_B$, we get the corresponding changed area $f_{c1}$ and unchanged area $f_{u1}$ from DI.

### 2.3. Morphological Reconstruction

After the images of the two areas have been acquired, the preconditions for sub-regional processing have been completed and there is still some noise in the image. However, FCM algorithm does not make use of the neighborhood information, and it is poor to noise. Therefore, the change-detection results in lots of scenes are not satisfactory. Morphological reconstruction [35] can reduce the noise of noisy images while preserving the target contour in FCM. Morphological reconstruction includes two basic operations: dilation and erosion. Suppose $f$ is the original image, and $P$ is the structuring element. The two basic morphological operators can be written as

$$f \oplus P(x,y) = \begin{array}{l} \sup f(x+a, y+b), \\ a,b \in D_b \\ x,y \in D_f \end{array} \tag{10}$$

$$f \ominus P(x,y) = \begin{array}{l} \inf f(x+a, y+b), \\ a,b \in D_b \\ x,y \in D_f \end{array} \tag{11}$$

where $f \oplus P(x,y)$ and $f \ominus P(x,y)$ are dilation and erosion operator of $f$ at pixel $(x,y)$, respectively. $D_b$ represents the domain of $P$, and $D_f$ represents the domain of $f$.

Through the combination of morphological dilation operator and erosion operator, some reconstruction operators with strong filtering ability can be obtained, such as morphological open operation and closed operation. They can be written as

$$f \circ P(x,y) = (f \ominus P) \oplus P(x,y) \tag{12}$$

$$f \bullet P(x,y) = (f \oplus P) \ominus P(x,y) \tag{13}$$

where $f \circ P(x,y)$ represents open operation, and $f \bullet P(x,y)$ represents closed operation. The open operation can reduce the noise of the image and remove the outliers in the image. The closed operation can fill the small cracks in the image without changing the position and size of the image block. Therefore, the alternative use of open operation and closed operation can remove the noise in the picture, while ensuring that the image is basically unchanged. In this paper, the morphological filtering of the image is achieved through open operation and closed operation, as shown in Equation (14).

$$F(x,y) = (f \circ P(x,y)) \bullet (P(x,y)) \tag{14}$$

where $F(x,y)$ represents the value of the pixel $(x,y)$ after morphological reconstruction.

In the above operations, the radius of the structuring element is 1. In paper [35], morphological reconstruction is only used in the original image. However, such operation is too rough for change detection. Therefore, the changed image is decomposed into three scales, and better results are obtained by morphological reconstruction of the three scales. They are shown by Equations (15)–(17).

$$f_{c2}(x,y) = (f_{c1} \circ n_1 P(x,y)) \bullet (n_1 P(x,y)) \tag{15}$$

$$f_{c3}(x,y) = (f_{c(1/2)} \circ n_1 P(x,y)) \bullet (n_1 P(x,y)) \tag{16}$$

$$f_{c4}(x,y) = (f_{c(1/4)} \circ n_1 P(x,y)) \bullet (n_1 P(x,y)) \tag{17}$$

where $f_{c1}$ is the original-scale image, $f_{c(1/2)}$ is the $\frac{1}{2}$-scale image, $f_{c(1/4)}$ is the $\frac{1}{4}$-scale image, and $n_1$ is the size of the structuring element. Different weight coefficients are used to fuse the three images. Then, the expression of the final changed image can be written as

$$f_c(x,y) = \alpha f_{c2}(x,y) + \beta f_{c3}(x,y) + \gamma f_{c4}(x,y) \tag{18}$$

For unchanged areas, it is more appropriate to use larger structuring elements for morphological reconstruction on the original scale image. It is written as

$$f_u(x, y) = (f_{u1} \circ n_2 P(x, y)) \bullet (n_2 P(x, y)) \tag{19}$$

Finally, $f_c$ and $f_u$ are summed to obtain the final morphological reconstruction image $\xi$.

$$\xi(x, y) = f_c(x, y) + f_u(x, y) \tag{20}$$

*2.4. Fuzzy C-Means Clustering*

FCM is a classical change-detection method. Its objective function is to find the fuzzy clustering of given data by minimizing the objective function. Its objective function can be calculated by Equation (21).

$$J_m = \sum_{l=1}^{q} \sum_{k=1}^{c} (u_{kl})^m \|y_l - v_k\|^2 = \sum_{l=1}^{q} \sum_{k=1}^{c} (u_{kl})^m d_{kl}^2 \tag{21}$$

where $Y = (y_1, y_2, \ldots, y_q)$ denotes a set of data samples, $V = (v_1, v_2, \ldots, v_k)$ denotes the clustering center of data, $U = [u_{kl}]_{q \times c}$ is the membership matrix of the samples, $u_{kl} \in [0, 1]$ is the degree of membership belonging to class $k$, $\|y_l - v_k\|^2$ is the Euclidean distance between the *k-th* cluster center and the *l-th* sample, and $m \in (1, \infty)$ is a weighted index. We introduce morphologically reconstructed images into clustering. Therefore, the objective function of the multi-scale morphological reconstruction fuzzy-c-means (MSMRFCM) clustering algorithm can be written as

$$J_m = \sum_{l=1}^{q} \sum_{k=1}^{c} \chi_l (u_{kl})^m \|\xi_l - v_k\|^2 \tag{22}$$

where $\chi_l$ denotes the number of pixels of *l-th* gray level, $\xi$ denotes the image after morphological reconstruction, $\xi_l$ means gray level, and $\|\xi_l - v_k\|^2$ is the Euclidean distance between the *k-th* cluster center and the *l-th* gray level. The optimal $u_{kl}$ and $v_k$ can be obtained by Lagrange multiplier method, which are as follows.

$$u_{kl} = \frac{\|\xi_l - v_k\|^{\frac{-2}{(m-1)}}}{\sum\limits_{j=1}^{c} \|\xi_l - v_j\|^{\frac{-2}{(m-1)}}} \tag{23}$$

$$v_k = \frac{\sum\limits_{i=1}^{q} \gamma_l (u_{kl})^m \xi_l}{\sum\limits_{i=1}^{q} \gamma_l (u_{kl})^m} \tag{24}$$

Each pixel is assigned to the corresponding class according to the final membership degree, and the change detection result is generated.

Many improved FCM algorithms, such as FLICM, introduce spatial local information into the objective function. This will greatly increase the computational complexity of the algorithm. By introducing morphological reconstruction into FCM, we not only reduce the computation of the algorithm, but also have good robustness to different kinds of noise. In addition, the introduction of multi-scale images can better obtain the complete features of the image. Finally, the running time of the algorithm is greatly reduced by using gray histogram instead of pixel-by-pixel calculation.

## 3. Experimental Results

### 3.1. Data Set

In this section, six real SAR image data sets are used to demonstrate the superiority of the method. Figures 2–7 show the SAR data sets. Figure 2 is the Bern data set. They were taken in July 1999 and May 1999 by ERS-2. The image size and resolution are 301 × 301 and 20 m. Figure 3 is the Ottawa data set. They were taken in May 1997 and August 1997 by Radarsat-1. The image size and resolution are 290 × 350 and 12 m. Figure 4 is the Farmland data set. They were taken in June 2008 and June 2009 by Radarsat-2. The image size and resolution are 306 × 291 and 8 m. Figure 5 is the Coastline data set. They were taken in June 2008 and June 2009 by Radarsat-2. The image size and resolution are 450 × 280 and 8 m. Figure 6 is the Inland Water data set. They were taken in June 2008 and June 2009 by Radarsat-2. The image size and resolution are 291 × 444 and 8 m. Figure 7 is the Bangladesh data set. They were taken in April 2007 and July 2007 by Envisat. The image size and resolution are 300 × 300 and 10 m. A description of these data sets is shown in Table 1.

**Table 1.** The 6 data sets used in the experiments.

| Place | Pre-Data | Post-Data | Size | Satellite | Resolution |
|---|---|---|---|---|---|
| Bern | 1999.04 | 1999.05 | 301 × 301 | ERS-2 | 20 m |
| Ottawa | 1997.05 | 1997.08 | 290 × 350 | Radarsat-1 | 12 m |
| Farmland | 2008.06 | 2009.06 | 306 × 291 | Radarsat-2 | 8 m |
| Coastline | 2008.06 | 2009.06 | 450 × 280 | Radarsat-2 | 8 m |
| Inland water | 2008.06 | 2009.06 | 291 × 444 | Radarsat-2 | 8 m |
| Bangladesh | 2007.04 | 2007.07 | 300 × 300 | Envisat | 10 m |

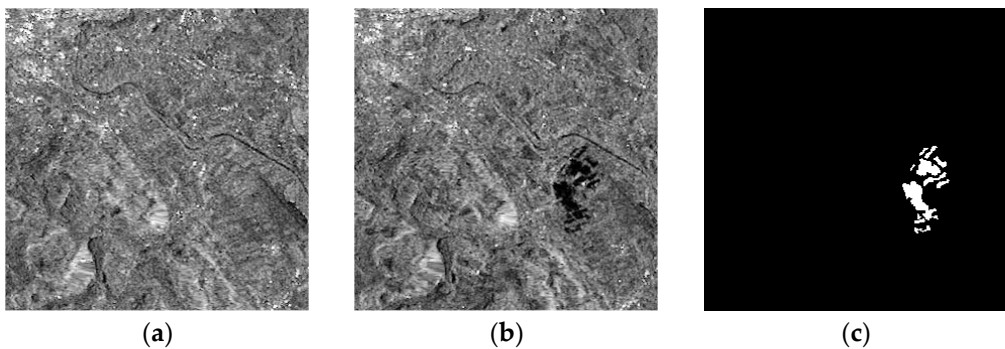

　　　　(a)　　　　　　　　　　　　　　　(b)　　　　　　　　　　　　　　　(c)

**Figure 2.** Bern data set. (**a**) Image obtained in April 1999; (**b**) image obtained in May 1999; (**c**) the ground truth.

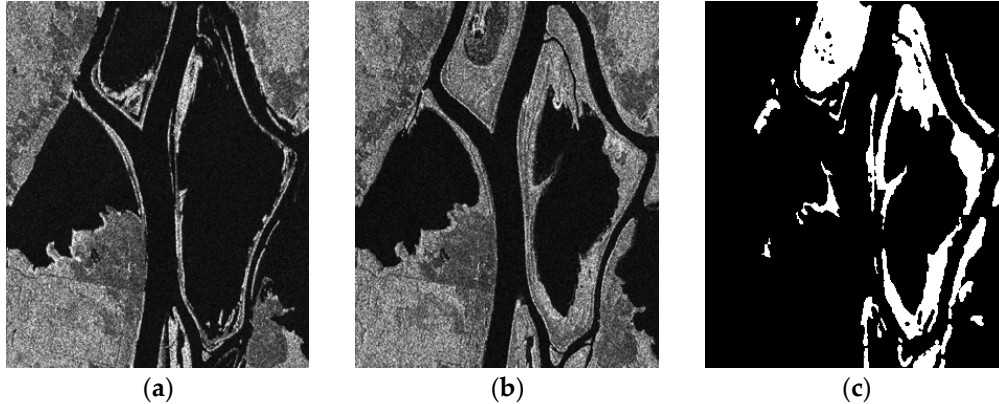

　　　　(a)　　　　　　　　　　　　　　　(b)　　　　　　　　　　　　　　　(c)

**Figure 3.** Ottawa data set. (**a**) Image obtained in May 1997; (**b**) image obtained in August 1997; (**c**) the ground truth.

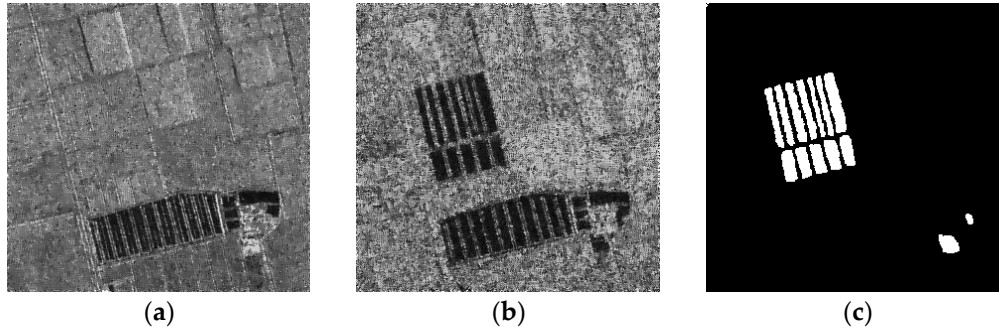

**Figure 4.** Farmland data set. (**a**) Image obtained in June 2008; (**b**) image obtained in June 2009; (**c**) the ground truth.

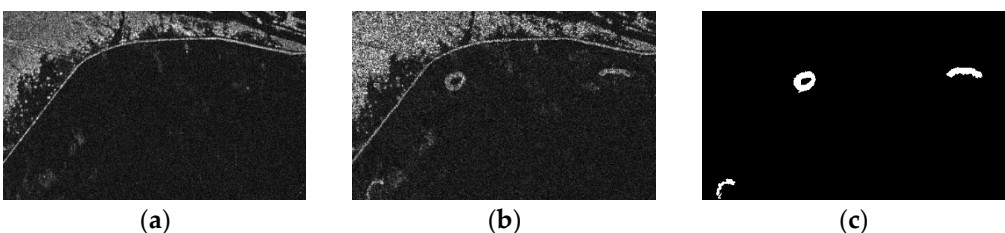

**Figure 5.** Coastline data set. (**a**) Image obtained in June 2008; (**b**) image obtained in June 2009; (**c**) the ground truth.

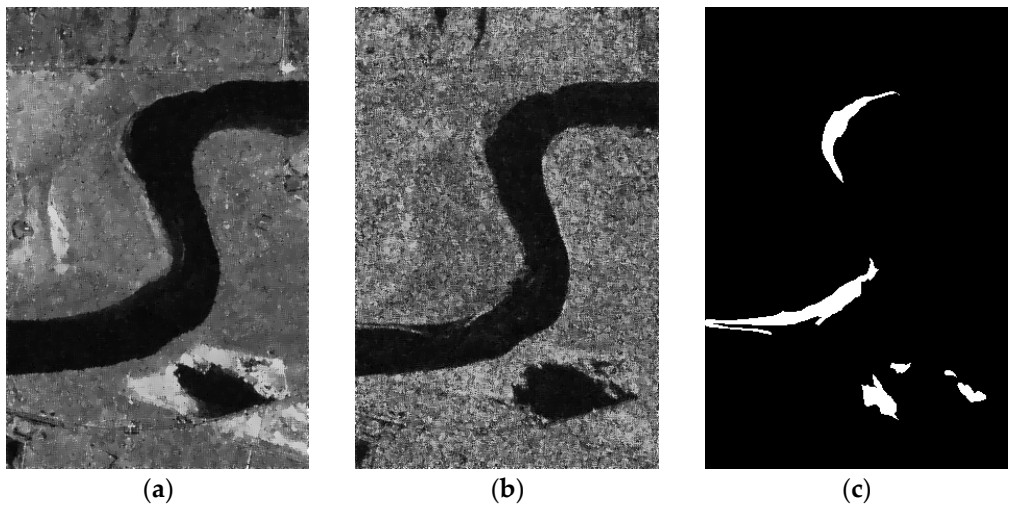

**Figure 6.** Inland Water data set. (**a**) Image obtained in June 2008; (**b**) image obtained in June 2009; (**c**) the ground truth.

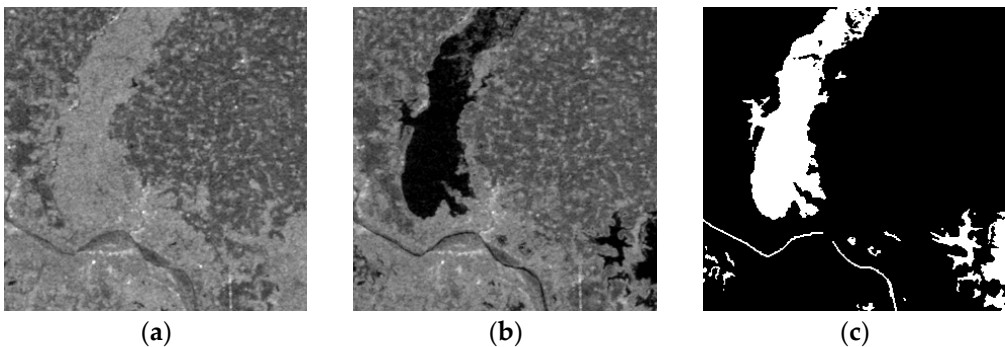

**Figure 7.** Bangladesh data set. (**a**) Image obtained in April 2007; (**b**) image obtained in July 2007; (**c**) the ground truth.

### 3.2. Evaluation Criterion

In order to more objectively explain the effect of change detection, we give five quantitative evaluation criterions of detection results: number of false negatives (*FN*), number of false positives (*FP*), number of overall errors (*OE*, the sum of *FP* and *FN*), percentage correct classification (*PCC*, the ratio of the number of correctly detected pixels to the total number of pixels) [47], and Kappa coefficient (*KC*, the similarity between the detection result image and the ground truth) [48]. The actual number of changed and unchanged pixels is $N_c$ and $N_u$. These indicators are calculated by the following.

$$OE = FP + FN \tag{25}$$

$$PCC = \frac{N_u + N_c - FP - FN}{N_u + N_c} \times 100\% \tag{26}$$

$$KC = \frac{PCC - PRE}{1 - PRE} \tag{27}$$

$$PRE = \frac{(N_c - FN + FP) \cdot N_c + (N_u - FP + FN) \cdot N_u}{(N_c + N_u) \cdot (N_c + N_u)} \tag{28}$$

### 3.3. DI Analysis

For DI analysis, we will take the Ottawa data set as an example. The images before and after change, saliency images, DIs, and change-detection results are shown in Figure 8. Figure 8a,b are the images before and after change, respectively. Figure 8c,d are the saliency image and binary image obtained by RMR DI, respectively. Figure 8e–h are R DI, MR DI, RMR DI, and SRMR-MSMR DI, respectively. Figure 8i–l are the change-detection results of these DIs, respectively. The evaluation indicators are shown in Table 2.

**Table 2.** Change-detection evaluation indicators of Ottawa data set by different DIs.

|          | FP   | FN   | OE   | PCC (%) | KC (%) |
|----------|------|------|------|---------|--------|
| R [16]   | 1631 | 1287 | 2918 | 97.13   | 89.29  |
| MR [18]  | 2323 | **193** | 2516 | 97.52   | 91.66  |
| RMR      | **427** | 941  | 1368 | 98.65   | 94.87  |
| MSMR     | 670  | 499  | **1169** | **98.85** | **95.69** |

It can be seen from the binary saliency image that the changed area of the image is not continuous. The binary image is the approximate changed area of the image, which has a larger changed area than the ground truth. Therefore, it lacks a large number of details of the image, but reduces the missed detection. There is a lot of speckle noise in R DI, which makes the detection result of the image poor. The calculation process of R DI only involves ratio operation, so it is very sensitive to noise. There are a large number of isolated pixels in the final change image, and the FP value reaches 1631. The FN value reaches 1287, which is the worst. Due to the mean filtering of MR DI, the changed area becomes significantly larger, and the final change image has the most FP pixels, reaching 2323. Thanks to the multiplication operation, the noise in the unchanged area is reduced in the RMR DI. The KC value of RMR reaches 94.87%, which is far higher than those of R and MR. From the binary saliency image, the area within the green rectangle is the changed area. Compared with RMR, SRMR-MSMR successfully eliminates the surrounding misdetected pixels, so the changed area is completely preserved. The area within the red rectangle is the unchanged area. SRMR-MSMR completely eliminates these misdetected pixels. It can be seen that small-size structuring elements are used for the changed area, which can not only maintain the details of the image but also eliminate noise. Large-size structuring elements are used for the unchanged area, which can completely eliminate noise. This is thanks to the correct guidance provided by the saliency detection for subregional processing.

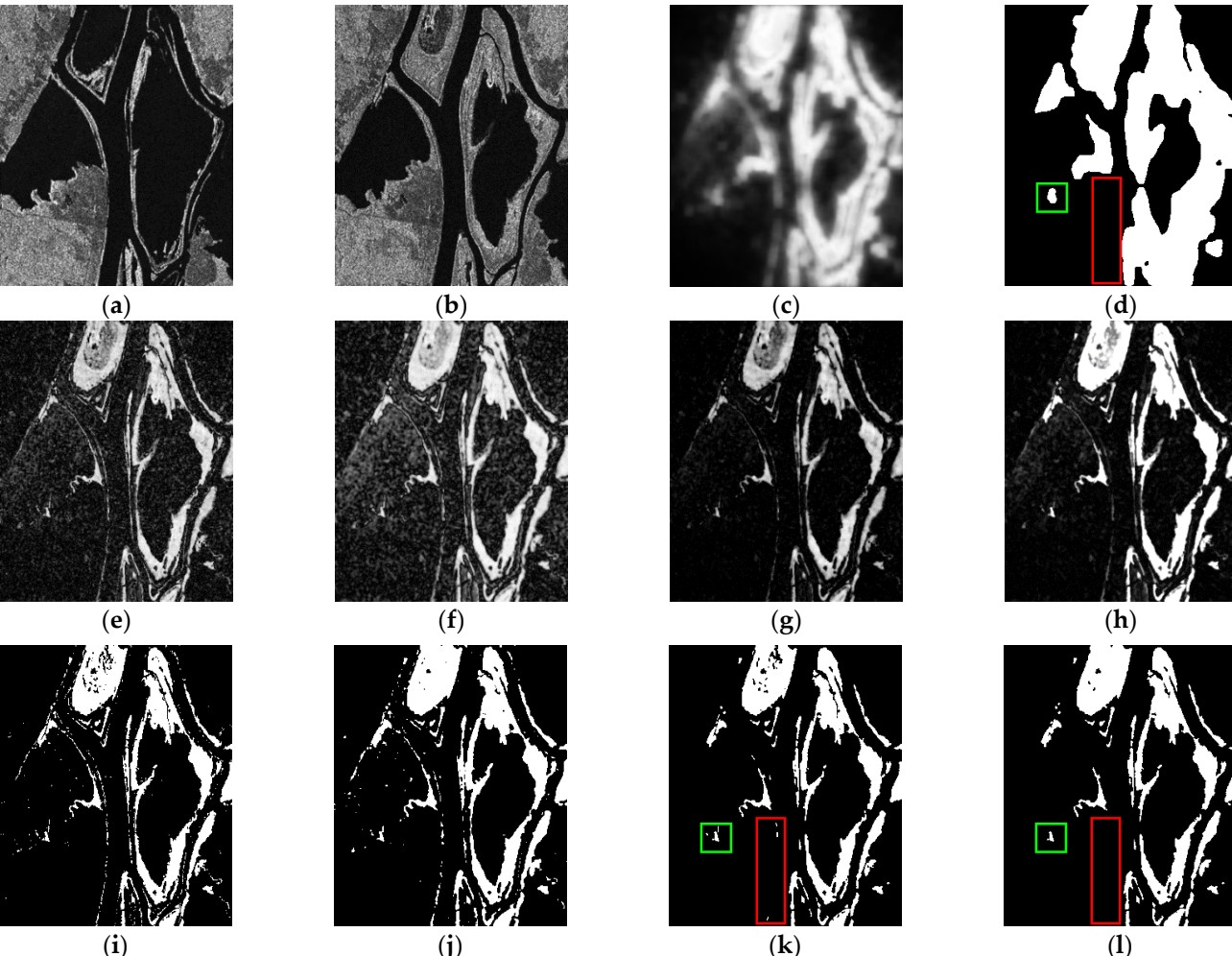

**Figure 8.** Images of the Ottawa data set. (**a**) Before; (**b**) after; (**c**) saliency image; (**d**) binary saliency image; (**e**) R; (**f**) MR; (**g**) RMR; (**h**) SRMR-MSMR; (**i**) result of R; (**j**) result of MR; (**k**) result of RMR; (**l**) result of SRMR-MSMR.

### 3.4. Change-Detection Results and Analysis

Six SAR image data sets and the change-detection results obtained by various methods are shown in Figures 9–14. The change-detection-result evaluation is shown in Tables 3–8. Eight methods are used as comparison methods for the proposed SRMR-MSMRFCM, which are FCM [29], FLICM [32], PCA-KMeans [28], PCANet [41], CWNN [40], MSAPNet [42], robust unsupervised small-area change detection (RUSACD) [21], and DDNet [43].

**Table 3.** Change-detection evaluation indicators of Bern data set of different methods.

|  | FP | FN | OE | PCC (%) | KC (%) |
|---|---|---|---|---|---|
| FCM [29] | 83 | 310 | 393 | 99.57 | 80.92 |
| FLICM [32] | 301 | **77** | 378 | 99.58 | 84.87 |
| PCA-KMeans [28] | 158 | 146 | 304 | 99.66 | 86.74 |
| PCANet [41] | **25** | 455 | 480 | 99.47 | 74.21 |
| CWNN [40] | 85 | 230 | 315 | 99.65 | 85.28 |
| MSAPNet [42] | 148 | 140 | 288 | **99.68** | 87.56 |
| RUSACD [21] | 307 | 122 | 429 | 99.53 | 82.57 |
| DDNet [43] | 71 | 246 | 317 | 99.65 | 84.98 |
| SRMR-MSMRFCM | 163 | 123 | **286** | **99.68** | **87.67** |

**Table 4.** Change-detection evaluation indicators of Ottawa data set of different methods.

|  | FP | FN | OE | PCC (%) | KC (%) |
|---|---|---|---|---|---|
| FCM [29] | 802 | 2139 | 2941 | 97.10 | 88.74 |
| FLICM [32] | 839 | 657 | 1496 | 98.53 | 94.49 |
| PCA-KMeans [28] | 970 | 1541 | 2511 | 97.53 | 90.57 |
| PCANet [41] | 871 | 1021 | 1892 | 98.14 | 92.97 |
| CWNN [40] | 1291 | 434 | 1725 | 98.30 | 93.75 |
| MSAPNet [42] | **262** | 2351 | 2613 | 97.43 | 89.79 |
| RUSACD [21] | 1468 | **295** | 1763 | 98.26 | 93.66 |
| DDNet [43] | 693 | 1010 | 1703 | 98.32 | 93.65 |
| SRMR-MSMRFCM | 670 | 499 | **1169** | **98.85** | **95.69** |

**Table 5.** Change-detection evaluation indicators of Farmland data set of different methods.

|  | FP | FN | OE | PCC (%) | KC (%) |
|---|---|---|---|---|---|
| FCM [29] | 2472 | 880 | 3352 | 96.23 | 70.39 |
| FLICM [32] | 1381 | **467** | 1848 | 97.92 | 82.76 |
| PCA-KMeans [28] | 1293 | 476 | 1769 | 98.01 | 83.37 |
| PCANet [41] | **25** | 1312 | 1337 | 98.50 | 84.77 |
| CWNN [40] | 324 | 734 | 1058 | 98.81 | 88.93 |
| MSAPNet [42] | 179 | 686 | 865 | 98.94 | 90.87 |
| RUSACD [21] | 124 | 1060 | 1184 | 98.67 | 86.98 |
| DDNet [43] | 231 | 855 | 1086 | 98.78 | 88.40 |
| SRMR-MSMRFCM | 102 | 709 | **811** | **99.01** | **91.35** |

**Table 6.** Change-detection evaluation indicators of Coastline data set of different methods.

|  | FP | FN | OE | PCC (%) | KC (%) |
|---|---|---|---|---|---|
| FCM [29] | 30,397 | 60 | 30,457 | 75.83 | 5.87 |
| FLICM [32] | 903 | 87 | 990 | 99.21 | 71.43 |
| PCA-KMeans [28] | 39,583 | 24 | 39,607 | 68.57 | 4.28 |
| PCANet [41] | 17,879 | **6** | 17,885 | 85.81 | 11.27 |
| CWNN [40] | 13,954 | 51 | 14,005 | 88.88 | 13.94 |
| MSAPNet [42] | 5794 | 58 | 5862 | 95.36 | 29.33 |
| RUSACD [21] | 115 | 174 | 289 | 99.77 | 88.92 |
| DDNet [43] | 144 | 184 | 328 | 99.74 | 87.52 |
| SRMR-MSMRFCM | **32** | 164 | **196** | **99.84** | **92.27** |

**Table 7.** Change-detection evaluation indicators of Inland Water data set of different methods.

|  | FP | FN | OE | PCC (%) | KC (%) |
|---|---|---|---|---|---|
| FCM [29] | 3268 | 543 | 3811 | 97.05 | 64.63 |
| FLICM [32] | 1654 | 798 | 2452 | 98.10 | 72.84 |
| PCA-KMeans [28] | 1354 | 603 | 1957 | 98.49 | 78.09 |
| PCANet [41] | **622** | 1770 | 2392 | 98.15 | 71.01 |
| CWNN [40] | 1333 | **494** | 1827 | 98.59 | 79.73 |
| MSAPNet [42] | 939 | 669 | **1608** | **98.76** | **81.04** |
| RUSACD [21] | 729 | 1114 | 1843 | 98.57 | 76.58 |
| DDNet [43] | 1334 | 576 | 1910 | 98.52 | 78.63 |
| SRMR-MSMRFCM | 721 | 912 | 1633 | 98.74 | 79.72 |

**Table 8.** Change-detection evaluation indicators of Bangladesh data set of different methods.

|  | FP | FN | OE | PCC (%) | KC (%) |
|---|---|---|---|---|---|
| FCM [29] | 4 | 4957 | 4961 | 94.49 | 74.27 |
| FLICM [32] | 21 | 3722 | 3743 | 95.84 | 81.43 |
| PCA-KMeans [28] | 308 | 4031 | 4339 | 95.18 | 78.46 |
| PCANet [41] | 5 | 4548 | 4553 | 94.94 | 76.73 |
| CWNN [40] | 19 | 3947 | 3966 | 95.59 | 80.17 |
| MSAPNet [42] | **2** | 4781 | 4783 | 94.69 | 75.35 |
| RUSACD [21] | 198 | 2886 | 3064 | 96.60 | 85.33 |
| DDNet [43] | 19 | 3774 | 3793 | 95.79 | 81.15 |
| SRMR-MSMRFCM | 139 | **2391** | **2530** | **97.19** | **88.05** |

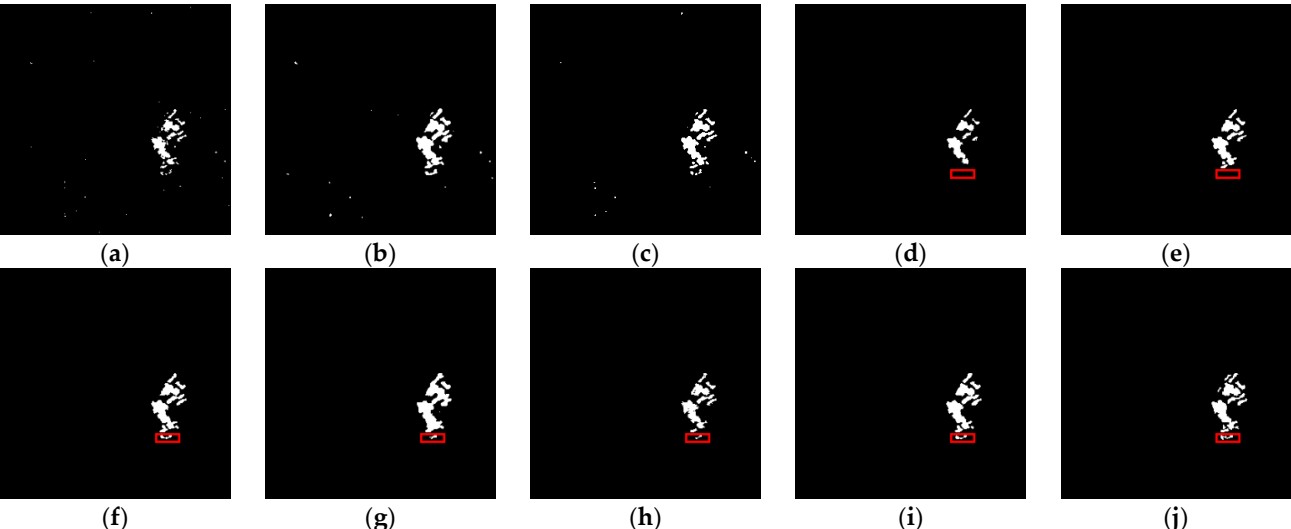

**Figure 9.** Change-detection results of the Bern data set. (**a**) FCM; (**b**) FLICM; (**c**) PCA-KMeans; (**d**) PCANet; (**e**) CWNN; (**f**) MSAPNet; (**g**) RUSACD; (**h**) DDNet; (**i**) SRMR-MSMRFCM; (**j**) the ground truth.

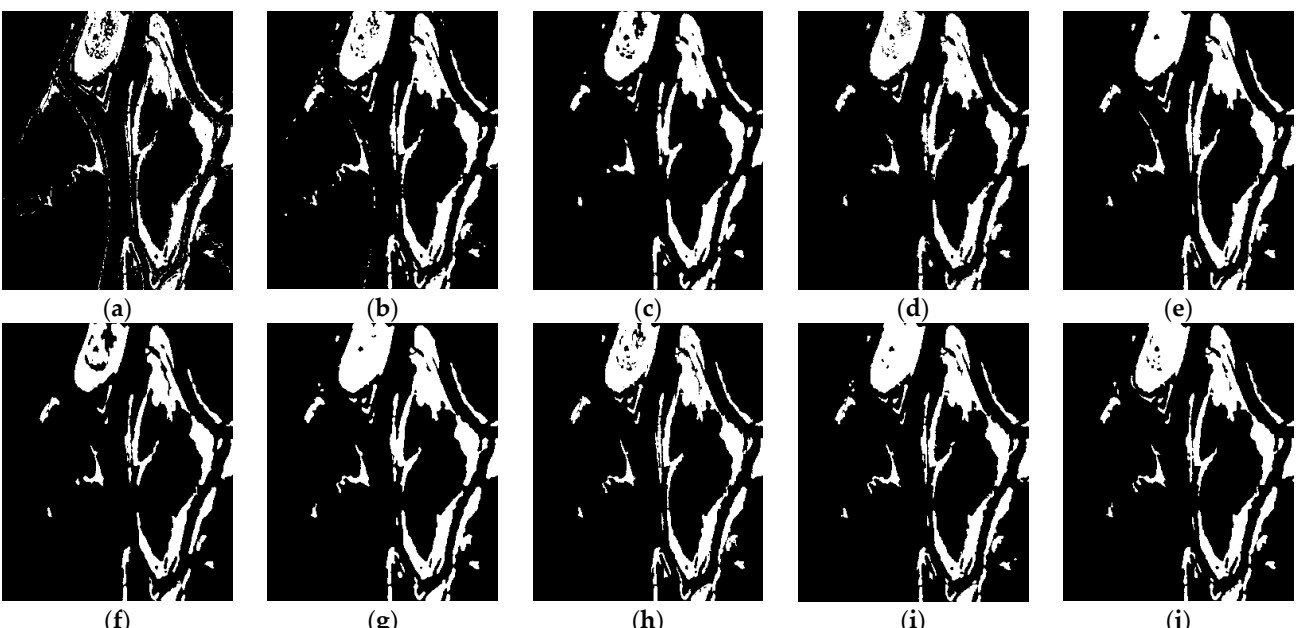

**Figure 10.** Change-detection results of the Ottawa data set. (**a**) FCM; (**b**) FLICM; (**c**) PCA-KMeans; (**d**) PCANet; (**e**) CWNN; (**f**) MSAPNet; (**g**) RUSACD; (**h**) DDNet; (**i**) SRMR-MSMRFCM; (**j**) the ground truth.

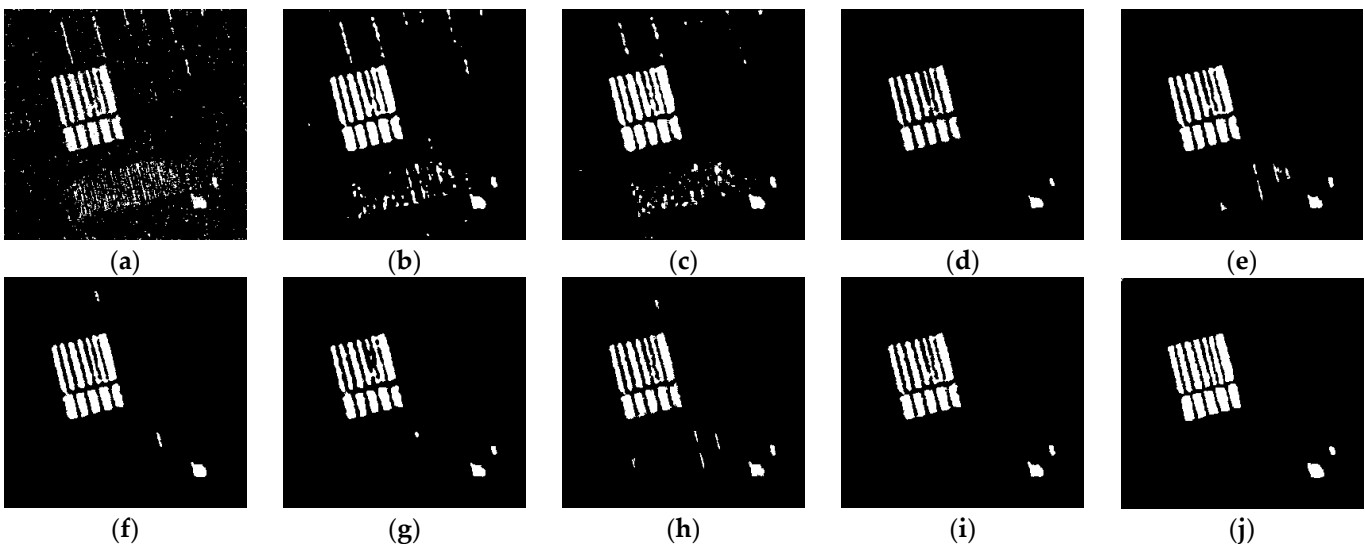

**Figure 11.** Change-detection results of the Farmland data set. (**a**) FCM; (**b**) FLICM; (**c**) PCA-KMeans; (**d**) PCANet; (**e**) CWNN; (**f**) MSAPNet; (**g**) RUSACD; (**h**) DDNet; (**i**) SRMR-MSMRFCM; (**j**) the ground truth.

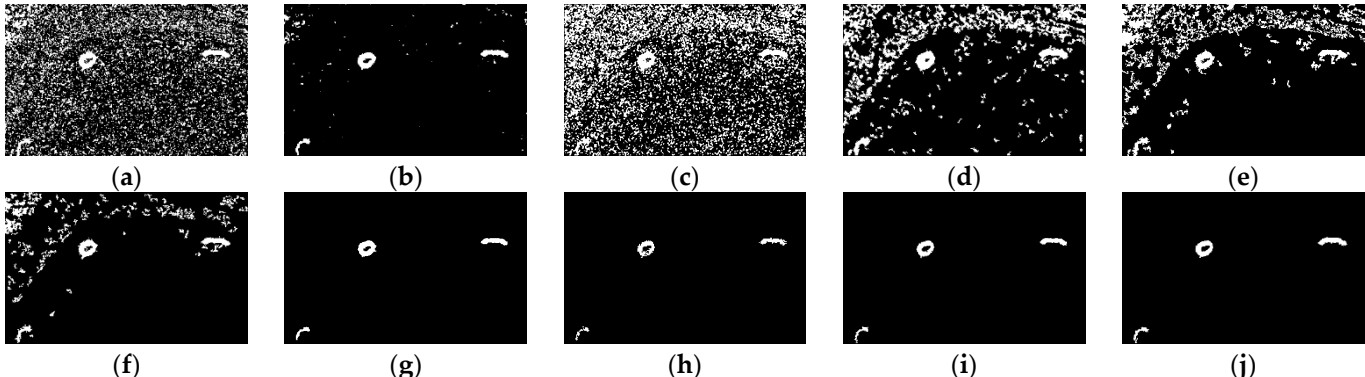

**Figure 12.** Change-detection results of the Coastline data set. (**a**) FCM; (**b**) FLICM; (**c**) PCA-KMeans; (**d**) PCANet; (**e**) CWNN; (**f**) MSAPNet; (**g**) RUSACD; (**h**) DDNet; (**i**) SRMR-MSMRFCM; (**j**) the ground truth.

The change-detection results and evaluation indicators of the Bern data set are shown in Figure 9 and Table 3, respectively. The change images generated by FCM, FLICM, and PCA-KMeans have many isolated pixels. Besides, the changed areas of FCM and DDNet are not continuous, resulting in a large number of FN values. The changed areas of FLICM and RUSACD are too large, resulting in a large number of FP values. The detection result of PCA-KMeans is very well, but there is too much noise. Inside the red rectangle, there are a lot of missing changed areas in PCANet, so the detection effect is poor. The result of CWNN is the same, but the situation is slightly better. Obviously, from the visual point of view, our method and MSAPNet have achieved the best detection results. Our method has a slight advantage over FN, while MSAPNet has a slight advantage over FP. There are no isolated pixels in the image, and the changed area is also kept completely. In terms of evaluation criteria, the KC value of the proposed SRMR-MSMRFCM is improved by 6.75%, 2.80%, 0.93%, 13.46%, 2.39%, 0.11%, 5.10%, and 2.69% over FCM, FLICM, PCA-KMeans, PCANet, CWNN, MSAPNet, RUSACD, and DDNet, respectively. Therefore, the proposed method has effective advantages in both visual and quantitative comparisons.

The change-detection results and evaluation indicators of the Ottawa data set are shown in Figure 10 and Table 4, respectively. Similar to the Bern data set, the existence of isolated pixels reduces the detection accuracy of FCM and FLICM. The edge of PCANet

is not smooth. Although the change image of CWNN is very smooth, a lot of image details are lost. Some small changed areas are not detected. Therefore, the edges remain poor. MSAPNet has a large number of FN pixels, while PCA-KMeans and RUSAD are the opposite. For this data set, the performance of DDNet is very ordinary, as the values of FP and FN are not outstanding. The proposed SRMR-MSMRFCM achieves the best detection results and effectively preserves the small changed area, while removing the isolated pixels. In terms of evaluation criteria, the KC value of the proposed SRMR-MSMRFCM is improved by 6.95%, 1.20%, 5.12%, 2.72%, 1.94%, 5.9%, 2.03%, and 2.04% over FCM, FLICM, PCA-KMeans, PCANet, CWNN, MSAPNet, RUSACD, and DDNet, respectively. Visually and metrically, the proposed method draws a balance between FP and FN.

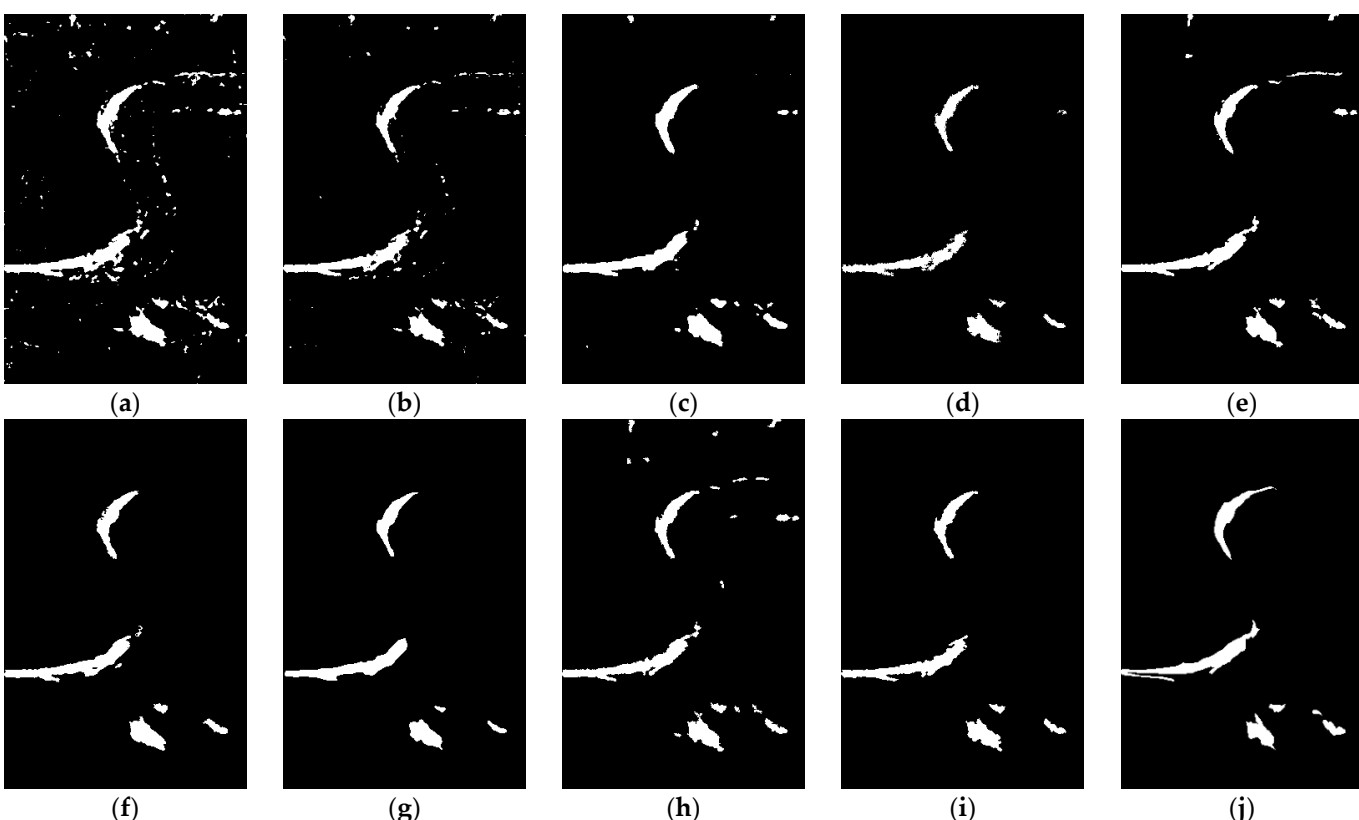

**Figure 13.** Change-detection results of the Inland Water data set. (**a**) FCM; (**b**) FLICM; (**c**) PCA-KMeans; (**d**) PCANet; (**e**) CWNN; (**f**) MSAPNet; (**g**) RUSACD; (**h**) DDNet; (**i**) SRMR-MSMRFCM; (**j**) the ground truth.

The change-detection results and evaluation indicators of the Farmland data set are shown in Figure 11 and Table 5, respectively. Since there is a large amount of noise in the original image, FCM, FLICM, and PCA-KMeans mistakenly judge the noise information as changed areas, resulting in a large number of FP values. There is almost no noise nor any isolated pixels in the image of PCANet and RUSACD, and the general changed areas and unchanged areas are detected. However, the detection results are not ideal because there are too many missed detection areas. CWNN and DDNet achieve better results in the changed area, but there are some false alarm areas. Our method has achieved excellent results in both changed and unchanged areas. MSAPNet has a similar performance to ours, but there are still some missed changed areas. It can be seen that our method has excellent robustness when the original image noise is too serious. In terms of evaluation criteria, the KC value of the proposed SRMR-MSMRFCM is improved by 20.96%, 8.59%, 7.98%, 6.58%, 2.42%, 0.48%, 4.37%, and 2.95% over FCM, FLICM, PCA-KMeans, PCANet, CWNN, MSAPNet, RUSACD, and DDNet, respectively. Therefore, the proposed method restores the changed areas with as little loss of information as possible.

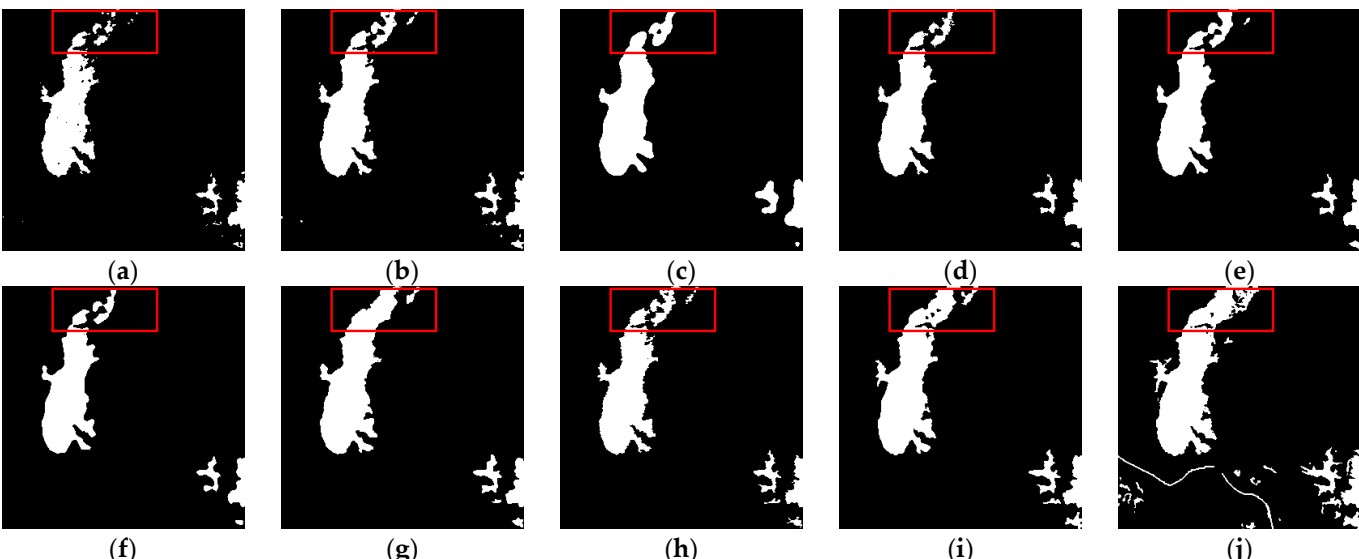

**Figure 14.** Change-detection results of the Bangladesh data set. (**a**) FCM; (**b**) FLICM; (**c**) PCA-KMeans; (**d**) PCANet; (**e**) CWNN; (**f**) MSAPNet; (**g**) RUSACD; (**h**) DDNet; (**i**) SRMR-MSMRFCM; (**j**) the ground truth.

The change-detection results and evaluation indicators of the Coastline data set are shown in Figure 12 and Table 6, respectively. For this data set, the detection results of FCM and PCA-KMeans are very poor. The FP value exceeds 30,000 and the error-detected pixels are almost all over the whole image. The results of PCANet, CWNN, and MSAPNet are better, but there are still a large number of block false detections. FLICM performs very well in this data set. The FP value is only 903, so the image noise is very small. RUSACD, DDNet, and our method all achieve excellent detection accuracy. By the naked eye, the change images of the three are almost the same as the ground truth. However, in the circular changed area, there are a small number of FP pixels in RUSACD and a small number of FN pixels in DDNet. From the evaluation criteria, our method achieves advantages in both FP and FN compared to those two methods. This is due to the fact that the MMR DI enlarges and reduces the gray levels of the changed area and the unchanged area, respectively. In addition, the MSMR algorithm effectively suppresses the noise. In terms of evaluation criteria, the KC value of the proposed SRMR-MSMRFCM is improved by 86.40%, 20.98%, 87.99%, 81.00%, 78.33%, 62.94%, 3.35% and 4.75% over FCM, FLICM, PCA-KMeans, PCANet, CWNN, MSAPNet, RUSACD and DDNet. For this data set, our method achieves much better results than the other methods, which proves its robustness.

The change-detection results and evaluation indicators of the Inland Water data set are shown in Figure 13 and Table 7, respectively. FCM, FLICM, PCAKMeans, and DDNet have the problem of a lot of FP pixels. PCANet and RUSACD have a large number of missed detections. CWNN and our method have their own advantages. In the red rectangle, CWNN has a large number of false alarm areas, but there is none in ours. CWNN has the advantage over FN value of 418, and we have the advantage over FP value of 612. Our PCC value is higher than CWNN by 0.15%, and the KC value is only 0.01% lower. MSAPNet achieves the best PCC and KC values. Although we do not achieve the best results for this data set, the noise in the change image is completely removed.

The change-detection results and evaluation indicators of the Bangladesh data set are shown in Figure 14 and Table 8, respectively. Obviously, the FP value of this data set is negligible, but there is a large number of missed detections, which can be seen from the image inside the red rectangle. The remaining methods, except RUSACD, all have FN values around 4000. RUSAD and the proposed SRMR-MSMRFCM successfully detected more changed areas. However, the proposed method leads by 59 and 495 pixels in FP and FN values, respectively. Thus, the proposed method effectively preserves the image details. In terms of evaluation criteria, the KC value of the proposed SRMR-MSMRFCM is

improved by 13.78%, 6.62%, 9.59%, 11.32%, 7.88%, 12.70%, 2.72%, and 6.90% over FCM, FLICM, PCA-KMeans, PCANet, CWNN, MSAPNet, RUSACD, and DDNet, respectively. It can be seen that the proposed method effectively reduces the missed detection.

For six real SAR image-change detection data sets, the proposed SRMR-MSMRFCM method achieves the best results for five of them. Obviously, the results of our method are much better than those of the classical methods, such as FCM, FLICM, and PCA-KMeans. Therefore, our analysis mainly focuses on the comparison with advanced deep learning methods and the mechanism of the methods.

Firstly, the multiplication operator effectively increases the contrast between the changed and unchanged areas. The R operator will improve the FN value of the detection result, while the MR operator will improve the FP value of the detection result. The reasons for these two problems are the ratio operation between pixels and the mean filtering of the neighborhood. In the FP area of the MR DI, the corresponding pixels have low gray values on the R DI. Therefore, the R operator can suppress the FP value of the MR operator after the multiplication operation. Similarly, in the FN area of the R DI, the corresponding pixels have higher gray values on the MR DI. Therefore, the MR operator can suppress the FN value of the R operator after the multiplication operation. Besides, compared with the fusion method with weighted summation, the method based on multiplication can amplify the change characteristics of the image. For data sets less affected by noise, such as the Bern and Bangladesh data sets, the comprehensive performance of the proposed method is much better than the deep learning methods due to the RMR DI. The proposed method detects the changed areas more completely, while most deep learning methods miss many changed areas. The reason is that these deep learning methods use LR DI to obtain labels. This leads to the omission of changed class pixel labels. Therefore, the neural network cannot fully learn the features of the changed areas, resulting in the high FN values.

Secondly, saliency detection and large-size structuring elements completely remove noise in unchanged areas. For the six data sets, there is almost no isolated noise in the detection of the proposed method. The CA method comprehensively considers the distance, mean value, and multi-scale information, so it completely detects the changed area of the image. Since the prior information of the two-dimensional Gaussian distribution matrix is introduced, the final saliency image can better reflect the change information of the image. Large-size structuring elements have strong denoising ability, but will destroy the details of the image. However, since they deal with unchanged areas, this disadvantage does not actually exist. The advantage of this method is reflected in the data sets that are heavily affected by noise, such as the Farmland, Coastline, and Inland Water data sets. It can be seen there are no isolated pixels in unchanged areas in the proposed method, but there are more or less for the other methods.

Thirdly, multi-scale images of changed areas enrich the features of the images and improve the detection accuracy. After the previous algorithm process, qualified results can be obtained by a single-scale image. However, in order to obtain better results, the method needs to be extended to multiple scales. The fusion of multi-scale images with appropriate proportion not only preserves the details, but also reduces the noise of the image. Small-size structuring elements play the same role here. The advantages of this method are also reflected in data sets that are heavily affected by noise, such as the Farmland, Coastline, and Inland Water data sets. It can be seen that the changed areas of the results are complete and smooth.

In order to prove the universality of the proposed method, four detection methods are used to test the simulation accuracy, which are manual threshold [49], Otsu, KMeans, and FCM. Neighborhood information and complex operations are not used in these methods, so they can be used for universality tests. The results of the Ottawa data set are shown in Figure 15 and Table 9. Obviously, for this data set, no matter what method is used, very similar and excellent detection results can be obtained. For all experiments, the values of PCC and KC are more than 98.80% and 95.50%, which are higher than those of the comparison methods in Section 3.4. It proves that the method has good universality.

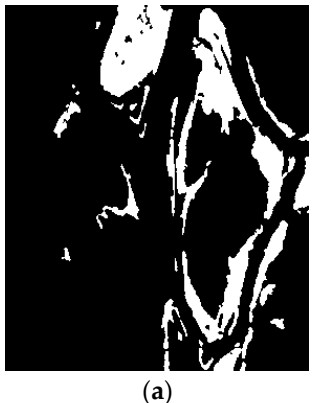 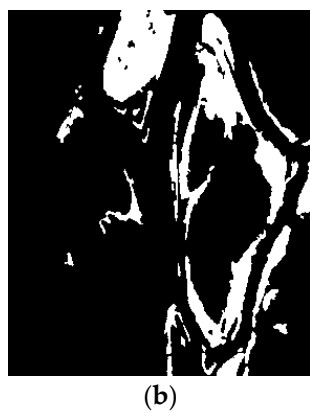 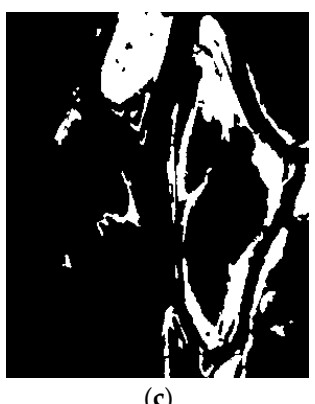 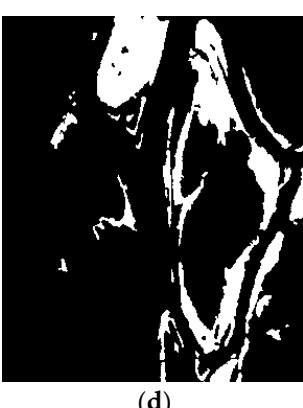

(**a**) (**b**) (**c**) (**d**)

**Figure 15.** Change-detection results of Ottawa data set of different methods. (**a**) threshold; (**b**) Otsu; (**c**) KMeans; (**d**) FCM.

**Table 9.** Change-detection results of Ottawa data set of different methods.

|  | FP | FN | OE | PCC (%) | KC (%) |
|---|---|---|---|---|---|
| Threshold [47] | **576** | 636 | 1212 | 98.81 | 95.51 |
| Otsu [26] | 626 | 566 | 1192 | 98.83 | 95.60 |
| KMeans [28] | 663 | 512 | 1175 | 98.84 | 95.67 |
| FCM [29] | 670 | **499** | **1169** | **98.85** | **95.69** |

## 4. Discussion

### 4.1. Discussion of Weight Coefficient

For parameter analysis, we will take the Ottawa data set as an example.

In order to prove the feasibility of MSMR, we designed five groups of experiments. The experimental setup is shown in Table 10. The results of the Ottawa data set are shown in Figure 16 and Table 10. In experiment A, only a 1/4-scale image is used. Accordingly, only a 1/2-scale image and the original image are used in experiments B and C, respectively. In experiment D, the images of all three scales occupy 1/3 of the proportion. The parameters for experiment E are those when the best results are obtained. As can be seen from the figure, in experiment A, when only the 1/4scale image is used, the detection image obtained loses a lot of details. The FP value is 744, and the KC value is only 89.89%. In experiment B, when only the 1/2–scale image is used, the image details are enriched and the accuracy improves a little. In experiment C, when only the original scale image is used, PCC and KC reach 98.60% and 94.60%, respectively, which are the second-best results. In experiment D, when the images of the three scales are used equally, the inspection accuracy is improved qualitatively, and KC reaches 94.43%. However, this value is slightly lower than that of experiment C, which only uses the original image. The reason is the wrong proportion of the three scales. There are rich details in the original image, so it should occupy the largest proportion. The smaller the scale of the image, the more information is lost, and there is less noise. Therefore, smaller-scale images should occupy smaller proportions. In experiment E, when the original image, the 1/2-scale image, and the 1/4-scale image occupy the proportions of 0.57, 0.32, and 0.08, respectively, we reach the highest PCC and KC value of 98.85% and 95.69%, respectively.

**Table 10.** Experimental parameter setting and change-detection results of Ottawa data set.

|   | $\alpha$ | $\beta$ | $\gamma$ | FP | FN | OE | PCC (%) | KC (%) |
|---|---|---|---|---|---|---|---|---|
| A | 0 | 0 | 1 | 744 | 1907 | 2651 | 97.39 | 89.89 |
| B | 0 | 1 | 0 | **62** | 1849 | 1911 | 98.12 | 92.59 |
| C | 1 | 0 | 0 | 210 | 1212 | 1422 | 98.60 | 94.60 |
| D | 1/3 | 1/3 | 1/3 | 314 | 1157 | 1474 | 98.55 | 94.43 |
| E | 0.57 | 0.32 | 0.08 | 670 | **499** | **1169** | **98.85** | **95.69** |

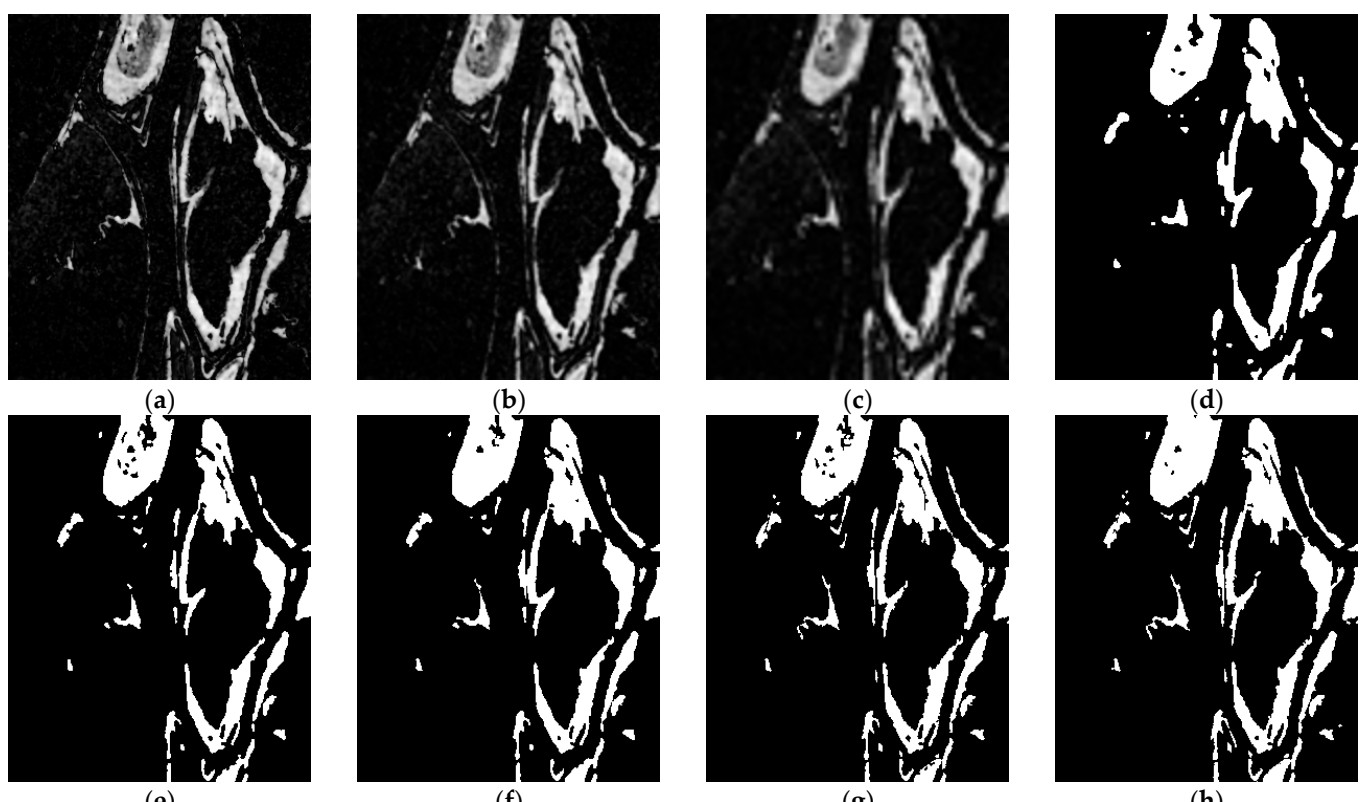

**Figure 16.** Multi-scale DI and change-detection results of the Ottawa data set. (**a**) Original DI; (**b**) 1/2-scale DI; (**c**) 1/4-scale DI; (**d**) result of A; (**e**) result of B; (**f**) result of C; (**g**) result of D; (**h**) result of E.

Figure 17 shows the results of all data sets. We can see that for single-scale images, the original images have achieved the best results, the second-best is the 1/2-scale, and the worst is the 1/4-scale. The results of experiment C were better than those of experiment D, except for the Farmland data set. This proves that the result of a multi-scale image with wrong proportions is not as good as that of the original image. The best results are all obtained in experiment E. For the Bern, Farmland, Coastline, Inland Water, and Bangladesh data sets, $\alpha$, $\beta$, and $\gamma$ are 0.5 0.4 0.1, 0.6 0.3 0.1, 0.38 0.31 0.29, 0.4 0.33 0.27, and 0.64 0.32 0.04, respectively.

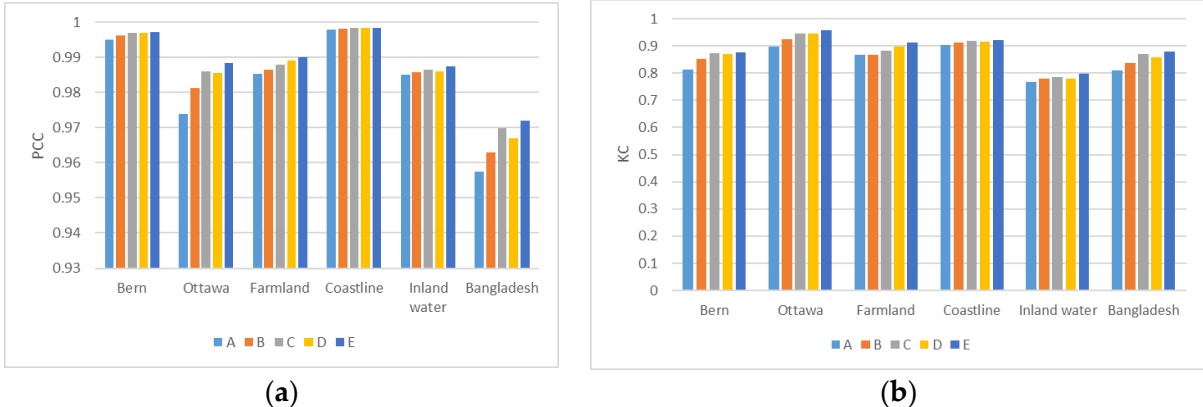

**Figure 17.** PCC and KC of all data sets. (**a**) PCC; (**b**) KC.

### 4.2. Discussion of Structuring Elements

In morphological reconstruction, the size of structuring elements is very important to the final result of image processing. This parameter will be analyzed by taking the Coastline data set as an example. The change-detection results and evaluation indicators of different-size structuring elements are shown in Figure 18 and Table 11, respectively. It can be seen that when the radius of the structuring element is 1, there are some isolated FP pixels in the image. When the radius is 2, there are very few FP pixels in the red rectangle. When the radii are 3 and 4, there are no isolated FP pixels in the image. The final detection results are almost the same as the ground truth. When the radius is 5, the changed area in the red rectangle is missed, and there are a lot of FP pixels in the green rectangle. When the radius is 6, the detection result is even worse, as the changed area in the blue rectangle is missed. It can be seen that if the structuring element is too small, the isolated noise may not be completely removed. If the structuring element is too large, both FP and FN pixels can seriously reduce the accuracy of detection. Therefore, it is necessary to select the appropriate size of structuring elements according to the noise level of the image. For the Coastline data set, the image noise is very serious, so the structuring element with radius 4 is chosen. For the Bern, Ottawa, Farmland, Inland Water, and Bangladesh data sets, the best radii are 1, 1, 3, 4, and 1, respectively.

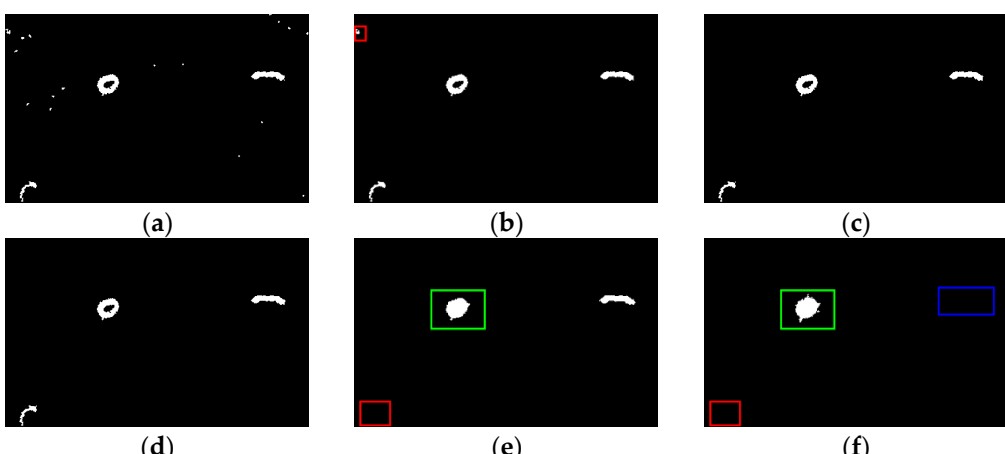

**Figure 18.** Change-detection results of different-size structuring elements of the Coastline data set. (**a**) Radius = 1; (**b**) radius = 2; (**c**) radius = 3; (**d**) radius = 4; (**e**) radius = 5; (**f**) radius = 6.

**Table 11.** Change-detection evaluation indicators of different-size structuring elements of Coastline data set.

|  | FP | FN | OE | PCC (%) | KC (%) |
|---|---|---|---|---|---|
| Radius = 1 | 178 | 196 | 374 | 99.70 | 85.88 |
| Radius = 2 | 63 | 182 | 245 | 99.81 | 90.39 |
| Radius = 3 | **29** | 176 | 205 | **99.84** | 91.88 |
| Radius = 4 | 32 | **164** | **196** | **99.84** | **92.27** |
| Radius = 5 | 176 | 307 | 483 | 99.62 | 80.98 |
| Radius = 6 | 239 | 729 | 968 | 99.23 | 55.75 |

## 5. Conclusions

In this paper, a new DI-generation method, RMR, is constructed by fusing the R DI and the MR DI based on multiplication. Experiments show that this method makes better use of the information of single pixels and neighborhood pixels. Therefore, it has excellent detection results for different kinds of SAR images. In addition, this paper proposes the MSMRFCM clustering change-detection method, based on multi-scale morphological reconstruction. Saliency detection is used to process images in different areas. Experiments show that the method is robust to noise and can maintain the details of the image. However, the method still has some shortcomings. There are four parameters in the method that can be adjusted. The size of the structuring elements can be easily adjusted to the best effect. However, although the coefficient adjustment of the three scale images is regular, it also takes some time to adjust. In future research, we will try to adapt or simplify the tuning of parameters without sacrificing too much accuracy. In addition, there is still much room for progress in the selection of the design of filters.

**Author Contributions:** Methodology, J.X.; validation, J.X.; software, Z.W. and Y.S.; writing—original draft preparation, J.X.; writing—review and editing, J.X. and Z.X.; supervision, Z.X. and G.L.; suggestions, P.H. All authors have read and agreed to the published version of the manuscript.

**Funding:** This research was funded by the National Natural Science Foundation of China under Grant No. 61801419 and the Natural Science Foundation of Yunnan Province under Grant Nos. 2019FD114 and 202201AT070027.

**Data Availability Statement:** Not applicable.

**Conflicts of Interest:** The authors declare no conflict of interest.

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
