# Peer review of "Change Detection Based on Fusion Difference Image and Multi-Scale Morphological Reconstruction for SAR Images"

_remotesensing, doi:10.3390/rs14153604_

Round 1

Reviewer 1 Report

This paper presents an efficient method for change detection in SAR images. This paper has a clear structure, a thorough discussion, and good experimental results, but there are a few problems:

1. About the abstract:

1-1 Clarify the fusion image in the abstract. (line 15);

1-2 What are the ratio(R) method and mean ratio(MR) method about (line 16);

1-3 neighborhood information is indistinct (line 21).

2. About the introduction:

2-1 For papers with more than two authors, please use et al. (line 42);

2-2 neighborhood information is indistinct (line 44);

2-3 Obtained→obtain (line 47).

3. About the formulas:

3-1 The symbols in the formula are consistent with the declared symbols: such as the normal body T_{1} (X, Y,) in formula (1)-(3), X_{d3} in formula (3), and {a} in formula (10);

3-2 The two different saliency values ​​(S_{i}) in formula (9) should be represented by different symbols;

3-3 There are formatting errors in formulas (21)-(23);

3-4 What is F(x,y) in formula (14);

3-5 f_{4}->f_{c4} in formula (18).

4. About the figures and tables:

4-1 The symbols in the formula can be added to the flowchart correspondingly (line 122);

4-2 The salient map in the flowchart comes from the fusion image obtained earlier, but formula (5) indicates that the salient map comes from the original image. In addition, images of Non saliency area and Saliency area are not only derived from Saliency image, so the arrows in the table are not correct (line122)

5. About the experiments:

5-1 The experimental results (FP) of PCAKmeans on Ottawa data are quite different from other papers (refer to Xiaofan Qu, Change Detection in Synthetic Aperture Radar Images Using a Dual-Domain Network);

5-2 Please add detection results based on the Fusion image (Fig. g in Figure 8) to illustrate the contributions of sec. 2.2-sec. 2.3.

Based on your experimental results, the proposed method does improve the final change detection accuracy. At the end of the paper, it is mentioned that your method still has shortcomings. Could you explain the shortcomings in more detail? The paper also contains some minor writing errors that need to be corrected.

Reviewer 2 Report

This manuscript utilizes a method based on fusion of difference images and multi-scale morphological reconstruction to finish the change detection for SAR image. The ratio method and the mean ratio method are combined to generate the difference image, which is a good idea. Saliency detection is also used to obtain the changed and unchanged areas of the image. And in the changed area, multi-scale morphological reconstruction is used to remove the noise. The structure is clear and the experiments are very rich. There are many analyses of the experiments but they are not deep enough. Overall, the ideas in this manuscript are interesting. However, there are some problems that need to be further considered. The specific opinions are as follows:

1.       On page 3, the third contribution (“FCM, Kmeans, Otsu, and manual threshold are applied to our method, and good results are obtained”) may be not accurate because these methods have long been proposed by others, the difference between the work you do and the works others do should be clarified.

2.       ” in Eq. (18) should be “”, right?

3.       Is the mathematical expression of Euclidean distance in Eq. (21), Eq. (22) and Eq. (23) correct? Please check it carefully.

4.       In section 2.4, the introduction of traditional method FCM takes up too much in this part, without highlighting the innovative changes or improvement you have made based on FCM. The introduction of your innovative ideas should be emphasized.

5.       In the analysis of experimental results, you have made a lot of comparative analysis of the experimental results themselves and clearly clarified the advantages of your method, but why not think about the reasons behind the experimental results and add them to your work?

6.       The latest comparison method selected in the article is proposed in 2019, why not use the lately proposed change detection method in recent years as the comparison method?

Reviewer 3 Report

This paper deals with the change detection for SAR images through fusion difference image and multi-scale morphological reconstruction, which can achieve relatively good detection performance. The experiments using six real SAR image data sets demonstrate the efficiency of the method. There are some questions which need to be explained. The specific questions are as follows.

1. In Section 2.1, LR operator is a more commonly used DI method, which can non-linearly map image change information better. Why are R operators and MR operators used to fuse rather than LR operators and MR operators?

2. In Section 2.3, why are larger structuring elements used in unchanged areas?

3. In Section 2.4, Equation (22) has additional parameters chi_l and ksi_l compared with Equation (21). What are the meanings of these parameters?

4. The paper lacks the analysis of structuring elements size in the experimental results. Please add analysis of this parameter.

5. In Figure 17, The histogram is lack of ordinate information. Please modify it.

6. Check the English written carefully, such as ‘‘change area’’.

Round 2

Reviewer 2 Report

Compared with the previous manuscript, the revised manuscript has been greatly improved, which can be found in the following aspects. First, some mistakes in grammar, variable representation and formulas have been corrected, and the structure of the revised manuscript is much clearer. Second, the innovation and the advantages of the proposed method in this manuscript are clearer than that previously stated, especially in the summary of introduction section and the introduction of the proposed method. Additionally, the analysis of the experimental results is more in-depth than before, which reflects the author's conscientious thinking about the reasons behind the experimental results. Finally, in terms of comparative experiments, three methods which are proposed more recently are added to the comparative experiment, further reflecting the superiority of the method proposed in this manuscript. And it makes the experimental richer and the experimental results more convincing. Therefore, it is suggested that this manuscript could be accepted.